# Magnesium Oxide in Constipation

**DOI:** 10.3390/nu13020421

**Published:** 2021-01-28

**Authors:** Hideki Mori, Jan Tack, Hidekazu Suzuki

**Affiliations:** 1Translational Research Center for Gastrointestinal Diseases (TARGID), University of Leuven, 3000 Leuven, Belgium; koyamaru2002@yahoo.co.jp (H.M.); jan.tack@kuleuven.be (J.T.); 2Division of Gastroenterology and Hepatology, Department of Internal Medicine, Tokai University School of Medicine, Isehara 259-1193, Japan

**Keywords:** magnesium oxide, constipation, laxative, hypermagnesemia

## Abstract

Magnesium oxide has been widely used as a laxative for many years in East Asia, yet its prescription has largely been based on empirical knowledge. In recent years, several new laxatives have been developed, which has led to a resurgence in interest and increased scientific evidence surrounding the use of magnesium oxide, which is convenient to administer, of low cost, and safe. Despite these advantages, emerging clinical evidence indicates that the use of magnesium oxide should take account of the most appropriate dose, the serum concentration, drug–drug interactions, and the potential for side effects, especially in the elderly and in patients with renal impairment. The aim of this review is to evaluate the evidence base for the clinical use of magnesium oxide for treating constipation and provide a pragmatic guide to its advantages and disadvantages.

## 1. Introduction

Magnesium began to be used medicinally in Western countries after the 1618 discovery by a farmer in Epsom, England that well water had a healing effect on cattle skin diseases [1]. The substance, magnesium sulfate, MgSO_4_, became known as Epsom salts and was used as a treatment for constipation for the next 350 years. However, the history of the medicinal use of magnesium in Eastern countries is even older. Magnesium nitrate, Mg(NO_3_)_2_, was already being used in Chinese herbal medicine when magnesium sulfate was discovered in the West, and magnesium nitrate was introduced to Japan in the 8th century [2]. The expected laxative actions of magnesium nitrate meant it was used to treat constipation alongside rhubarb, which contains an anthraquinone glycoside compound that acts as a stimulant laxative in Chinese and Japanese herbal medicine [3]. Magnesium oxide was introduced from the West to the East in the 19th century when German physician P.F.B. Siebold brought magnesium oxide to Japan [4,5]. Following this, magnesium oxide took a central position as a laxative of choice in East Asian countries such as Japan, China, and Taiwan [6,7]. On the other hand, the magnesium preparation most commonly used in South Korea and the United States is magnesium hydroxide [8,9]. In European countries, magnesium hydroxide, magnesium citrate, magnesium sulfate, and magnesium oxide are used as saline laxatives, but there are only a limited number of studies that compare the different salt forms and few actual cases of use [10,11]. Therefore, at present, the decision of which magnesium salt to use as a laxative is dependent upon the country in which it is prescribed. Polyethylene glycol is the laxative of first choice in the United States [12] and is also widely used in Europe [10]. In recent years, new drugs such as a type-2 chloride channel activator, a guanylate cyclase 2C receptor agonist, and an inhibitor of the ileal bile acid transporter have been developed to treat constipation and can be given to a patient sequentially [13,14,15]. While the range of laxatives available has expanded, there is still no consensus on how to use them correctly. Currently, the safety, convenience, and low cost of magnesium oxide, which has been used for many years, mean that it has once again attracted attention (Table 1).

In this article, we address the evidence for using magnesium oxide to treat constipation, and highlight the advantages and limitations that should guide clinical decisions about the use of this therapy.

## 2. Mechanism of Action

Magnesium oxide (MgO) is converted into magnesium chloride (MgCl_2_) under acidic conditions in the stomach. Thereafter, MgCl_2_ is converted to magnesium bicarbonate Mg(HCO_3_)_2_ by sodium hydrogen carbonate (NaHCO_3_) from pancreatic secretion in the duodenum, and finally becomes magnesium carbonate (MgCO_3_) (Figure 1). Mg(HCO_3_)_2_ and MgCO_3_ increase the osmotic pressure of the intestinal lumen fluid, thereby promoting the transfer of water to the intestinal lumen and increasing the water content and volume of the stool. In addition, the swollen stool stimulates the intestinal wall and intestinal propulsive motor activity.

Anthraquinone-based drugs, which act as stimulant laxatives and include rhubarb and senna, cause tolerance with continuous use, whereas patients do not develop tolerance to magnesium oxide with continuous use.

Although magnesium oxide use can lead to hypermagnesemia, this is rare. This, combined with a lack of pharmacokinetic information on magnesium oxide, means that in clinical practice it is often prescribed with little regard for the optimal dose. Siener et al. administered 404 mg/day of magnesium oxide to healthy volunteers and analyzed the magnesium concentration in blood and urine. There was no significant change in blood concentration of magnesium, but urinary magnesium excretion increased by 40% after administration of magnesium oxide [16]. Yoshimura et al. performed a pharmacokinetic study of orally administered magnesium oxide in rats [17]. They showed that 85% of magnesium is excreted in feces, while 15% of magnesium is absorbed from the intestinal tract and excreted in urine [17]. Furthermore, they showed that the plasma magnesium concentration in rats was maintained at a high level for a relatively long period due to the slow absorption of magnesium [17]. Such data have not yet been shown in humans. The kidney plays a central role in maintaining magnesium homeostasis through active reabsorption, which is influenced by the sodium load in the tubules and possibly by acid–base balance. The kidneys of individuals with a normal glomerular filtration rate (GFR) filter approximately 2000–2400 mg of magnesium per day [18]. Based on this information, the dose of magnesium oxide usually used for laxative purposes is considered to have a low propensity to cause hypermagnesemia, while magnesium oxide should be used carefully in patients with renal dysfunction.

## 3. Evidence Underpinning the Use of Magnesium Oxide

### 3.1. Functional Constipation in Adults

Magnesium oxide has long been used to treat constipation, and an estimated 10 million patients in Japan are treated with this agent annually [19]. Although the prescription of magnesium oxide has been based on empirical evidence for many years, two randomized controlled trials (RCTs) showing the effectiveness of magnesium oxide for treating chronic constipation in adults have recently been reported. Mori S et al. conducted a randomized, double-blind placebo-controlled study comparing magnesium oxide with placebo [20]. In this study, magnesium oxide administration led to superior overall improvement of symptoms, and improved spontaneous bowel movement, stool form, colonic transit time, abdominal symptom, and quality of life. Patients treated with magnesium oxide had a response rate of 70.6% for overall symptom improvement, which was significantly higher than the response rate of 25.0% observed in the placebo group [20]. Morishita et al. conducted a randomized, double-blind placebo-controlled study comparing magnesium oxide with placebo and senna [21]. The response rate for overall improvement was 11.7% in the placebo group, 69.2% in the senna group, and 68.3% in the magnesium oxide group. Patients receiving senna or magnesium oxide had significant improvements in spontaneous bowel movement and constipation-related quality of life, compared with patients in the placebo group. Moreover, no severe treatment-related adverse events were observed in either treatment group. The time to the first spontaneous bowel movement in this study was significantly shortened for the senna group (18.8 h) and magnesium oxide group (17.9 h) compared with the placebo group (22.0 h). However, the authors discussed that senna and magnesium oxide were less effective in reducing time to first spontaneous bowel movement than newer drugs: lubiprostone (3.5 h versus 48.0 h for placebo), linaclotide (6.7 h versus 24.7 h for placebo), and elobixibat (5.1 h versus 25.5 h for placebo) [21]. These differences may serve as a reference for the correct use of these drugs for treating constipation in the future.

### 3.2. Opioid-Induced Constipation

Opioid-induced constipation is the most common and problematic complication of opioid therapy [22,23,24]. Standard laxatives, such as osmotic agents (macrogol) and stimulants (bisacodyl, picosulfate, and senna), are good first-line choices in the management of opioid-induced constipation [22]. Second-line agents that block μ-opioid receptors in the gastrointestinal tract but do not enter the central nervous system, so called PAMORAs (peripherally acting mu opioid receptor antagonists), can be used to treat opioid-induced constipation without diminishing central analgesic actions [22]. The mu-opioid receptor antagonists methylnaltrexone, naloxegol, and naldemedine are safe and effective treatments for opioid-induced constipation [22,25,26,27,28,29]. Evidence is not yet available on the use of magnesium oxide for treating opioid-induced constipation; however a single-institution, open-label, randomized controlled trial comparing the effectiveness of magnesium oxide with naldemedine for preventing opioid-induced constipation is ongoing [30]. Magnesium oxide could be a promising drug candidate for opioid-induced constipation, yet patients taking opioids often take gastric antisecretory drugs such as proton pump inhibitors and histamine H_2_ receptor antagonists, which can diminish the efficacy of magnesium oxide [31,32]. Moreover, retrospective studies have shown that 94% of opioid-induced constipation patients did not achieve an inadequate response to one laxative agent with one laxative treatment including magnesium hydrochloride [33]. On the other hand, PAMORAs may be advantageous, as opioids act on receptors in both the small and large intestine and, second, dampen neuronal activity also in the gut [22].

### 3.3. Functional Constipation in Children

Magnesium salts are also used for treatment of functional constipation in children [34]. Two RCTs showing the effectiveness of magnesium oxide for treating chronic constipation in children have been reported. Bu et al. conducted a double-blind placebo-controlled, randomized study to compare *Lactobacillus casei rhamnosus Lcr35* with magnesium oxide and placebo [35]. The patients who received magnesium oxide or the probiotic had a higher defecation frequency, higher percentage of treatment success, less use of glycerin enema, and softer stools than the placebo group. There were no significant differences in the aforementioned comparisons between the magnesium oxide and probiotic groups; however, the onset of effects occurred slightly earlier in patients treated with magnesium oxide than those treated with probiotics [35]. Kubota et al. conducted a double-blind, placebo-controlled, randomized trial to compare the probiotic *Lactobacillus reuteri* DSM 17938 and magnesium oxide for relieving chronic functional constipation in children [36]. They divided the subjects into three groups: the first group received *L. reuteri* DSM 17938, the second group received *L. reuteri* DSM 17938 and magnesium oxide, and the third group received magnesium oxide. There was a significant improvement in the defecation frequency in all groups at the fourth week after treatment compared to baseline. In this study, the authors also investigated the relationship between gut microbiome composition, magnesium oxide, and defecation frequency. They showed that defecation frequency was higher in magnesium oxide-treated patients than in patients whose gut microbiome contained bacteria of the genus *Dialister*, and that defecation frequency negatively correlated with the frequency of bacteria belonging to the genus *Clostridiales* in patients’ gut microbiomes [36]. This result suggests that magnesium oxide treatment alters the gut microbiome. Although this is a noteworthy finding, the long-term health effects of an altered gut microbiome induced by magnesium oxide are unclear, and further research is likely necessary.

### 3.4. Guidelines on the Use of Magnesium Precautions for Functional Constipation

Guidelines for using magnesium preparations in various countries and regions are shown in Table 2. Magnesium oxide is mentioned only in Japanese guidelines, in which the recommendation level is “strong” [37]. Although sufficient international evidence of the use of magnesium oxide in adults only recently became available, the experience gained from prescribing magnesium oxide to more than 10 million patients annually was only acknowledged in the Japanese guidelines. Magnesium salts, including magnesium hydroxide, are recommended in the guidelines of the American Gastroenterological Association, the Asian Neurogastroenterology and Motility Association, the Korean Society of Neurogastroenterology and Motility, the Mexican Association of Gastroenterology, the Italian Association of Hospital Gastroenterologists and the Italian Society of Colorectal Surgery, and the French Society of Gastroenterology [38,39,40,41,42,43]. However, due to low levels of evidence, the recommendation level in these guidelines is weak. On the other hand, the German Society for Digestive and Metabolic Diseases, the German Society for Neurogastroenterology and Motility, and the German Society for Internal Medicine and the American College of Gastroenterology recommend against the use of magnesium hydroxide because of possible adverse effects [44,45]. The European Society of Neurogastroenterology and Motility, the UK’s National Institute for Health and Clinical Excellence, and an expert panel in Hong Kong do not mention magnesium preparations [46,47,48].

## 4. Practical Use of Magnesium Oxide

### 4.1. Dosage and Administration

Magnesium oxide is an osmotic laxative, and its key effect is a softening of hard stools; therefore, it is important to first ask the patient about the hardness of stools and the frequency of bowel movement. In real-life clinical practice, evaluation using the Bristol scale is useful as an objective index [49]. To choose a proper drug, a diagnostic process, including several clinical/psychiatric parameters, is also important [50,51].

The package insert of magnesium oxide advises: “In general, for adults, take 2 g of the active ingredient in 3 divided doses a day before or after meals, or once before bedtime” [52]. However, in practice, 2 g per day can result in hypermagnesemia; therefore, we recommend that a starting dose of approximately 1 g taken as two or three divided doses a day is used, and adjusted appropriately according to symptoms [6]. While there are cases in which 250 mg a day is sufficiently effective, there are rare cases in which sufficient improvements cannot be obtained even with a dose of 2 g a day [6].

If magnesium oxide alone is not effective, other laxatives such as stimulant laxatives, polyethylene glycol, lubiprostone, linaclotide, and elobixibat can also be used as adjunct drugs, in which case magnesium oxide should not be used in excess to avoid hypermagnesemia.

### 4.2. Drug Interactions

Magnesium oxide has an adsorptive action and an antacid action, and so it affects the absorption and excretion of other drugs. Tetracycline, new quinolones, and bisphosphonates may form chelates with magnesium, which diminishes the effects of these drugs. Therefore, there should be a sufficient time interval between dosing if these drugs are prescribed together. Considering the digestion time in the stomach, an interval of at least 2 h is recommended [53]. The effects of iron supplements, digitalis, polycarbophil calcium, and fexofenadine may be diminished by the adsorption action of magnesium or the magnesium oxide-induced increase the intragastric pH. The effects of cation-exchange resins may be decreased because magnesium ions exchange with the cations of these drugs. The effects of some cephem antibiotics, mycophenolate mofetil, delavirdine, zalcitabine, penicillamine, azithromycin, celecoxib, rosuvastatin, rabeprazole, and gabapentin, may be diminished by magnesium; the reasons for this reduced efficacy is not understood. Activated vitamin D supplements may cause hypermagnesemia because they can promote gastrointestinal absorption and reabsorption of magnesium from renal tubules. Consumption of large amounts of milk and calcium supplements may cause hypercalcemia and alkalosis due to increased renal reabsorption of calcium (known as milk–alkali syndrome). Magnesium oxide, an absorbable alkaline preparation, exacerbates calcium retention in the kidneys. The laxative effect of magnesium oxide is decreased in patients receiving a H_2_ receptor antagonist or a proton pump inhibitor due to the low solubility of magnesium oxide at the higher gastric pH and lower generation of MgCl_2_ and Mg(HCO_3_)_2_ [31,32].

### 4.3. Side Effect Profile and Toxicity

#### 4.3.1. Hypermagnesemia

The poor bioavailability of magnesium oxide makes it relatively safe, but prolonged treatment may induce hypermagnesemia [54]. In recent years, cases of magnesium oxide-induced hypermagnesemia resulting in serious outcomes have been reported [55,56,57,58]. Blood magnesium levels are usually tightly controlled by the kidneys; the normal range is 1.8–2.4 mg/dL, and levels 3.0 mg/dL and above are defined as hypermagnesemia. Serum magnesium concentrations >5.0 mg/dL have been associated with nausea, headache, light-headedness, and cutaneous flushing, and levels above 12 mg/dL have been associated with respiratory failure, complete heart blockage, and cardiac arrest [59]. Recently, the Japanese Ministry of Health, Labour and Welfare recommended that serum magnesium concentrations be measured periodically in geriatric patients and in patients administered magnesium oxide for prolonged periods [60].

Due to a lack of convincing evidence of the degree of risk of using magnesium salts in clinical practice, the authors of this review conducted a retrospective study on the occurrence of hypermagnesemia in patients receiving oral magnesium oxide for treating constipation [6]. Among the patients evaluated for serum magnesium concentration, 5.2% had hypermagnesemia and 16.6% had high serum magnesium concentration (>2.5 mg/dL). Factors associated with hypermagnesemia were impaired renal function and higher magnesium oxide dosage. Patients with a renal function classification of G1 (GFR ≥ 90 mL/min/1.73 m^2^) and G2 (GFR 60–89 mL/min/1.73 m^2^) had a serum magnesium concentration of 2.06 ± 0.23 and 2.11 ± 030 mg/dL, respectively, which is within the normal range. Patients classified as G4 (GFR 15–29 mL/min/1.73 m^2^) and G5 (GFR < 15 mL/min/1.73 m^2^) had a serum magnesium concentration of 2.46 ± 0.58 and 2.60 ± 0.99 mg/dL respectively; these averages exceeded the upper limit of normal levels. Analysis showed a significant positive correlation between the daily dose of magnesium oxide and blood magnesium concentration. By contrast, age and duration of administration were not correlated with serum magnesium concentration. From these results, we clarified that individuals with decreased renal function and individuals receiving a large daily dose are at high-risk of developing the hypermagnesemia. Wakai et al. also identified risk factors for developing hypermagnesemia in patients prescribed magnesium oxide via a retrospective cohort study [61]. They showed that 23% developed high serum magnesium concentration (>2.5 mg/dL). Renal function, daily dose, and duration of administration were indicated to be independent risk factors. Horibata et al. examined the relationship between renal function and serum magnesium concentration in elderly patients treated with magnesium oxide [62], and found that renal function also significantly correlated with serum magnesium levels. Tatsuki et al. investigated whether children with functional constipation taking daily magnesium oxide develop hypermagnesemia [63]. They showed that the serum magnesium concentration was 2.4 (2.3–2.5 median and interquartile range) mg/dL in children with functional constipation taking daily magnesium oxide, which was significantly higher than in the age- and sex-matched control group (2.2; 2.0–2.2 mg/dL). However, none of the patients had side effects associated with hypermagnesemia. These reports indicated the importance of monitoring serum magnesium levels in patients being treated with magnesium, especially in patients with chronic kidney failure and in patients treated with high dosages of magnesium oxide (Figure 2a).

Magnesium oxide has relatively poor bioavailability (a fractional absorption of 4%) compared to other magnesium salts; the fractional absorption of magnesium chloride, magnesium lactate, and magnesium aspartate are all between 9 and 11% [54]. Another report showed that the bioavailability of magnesium oxide was significantly lower than that of magnesium citrate [64]. The low bioavailability of magnesium oxide may be related to its low solubility in water [64]. These results suggest that magnesium oxide may have a less propensity to cause hypermagnesemia than other magnesium preparations.

#### 4.3.2. Milk–Alkali Syndrome

Milk–alkali syndrome is characterized by the triad of hypercalcemia, metabolic alkalosis, and decreased kidney function, and is caused by excessive intake of calcium and alkali [65]. Magnesium oxide is an absorbable alkaline preparation. Since serum calcium levels are tightly controlled by parathyroid hormone and active vitamin D, hypercalcemia does not easily occur even with excessive calcium intake [66]. However, when the serum calcium concentration rises (as a result of several possible causes) and the calcium concentration in the renal tubule rises, hypercalcemia has a well-known natriuretic and diuretic effect by activating the calcium-sensing receptor, leading to intravascular depletion of calcium. The resulting reduction in GFR further limits the excretion of bicarbonate and calcium, and absorbable alkaline preparations including magnesium oxide exacerbate calcium retention in the kidneys [67] (Figure 2b).

In recent years, common diseases in the aging population include both constipation and osteoporosis, which are often treated at the same time. In other words, individuals are often simultaneously treated with magnesium oxide for constipation and vitamin D preparations for osteoporosis, leading to milk–alkali syndrome [56,68,69]. Considering this scenario, it is necessary to pay attention to the possibility of milk–alkali syndrome and drug–drug interactions in the elderly who copresent with constipation and osteoporosis.

## 5. Summary and Future Perspectives

Magnesium oxide has been clinically used as a laxative for many years. Due to a lack of alternative treatment options, it was prescribed based on empirical experience. The increasing availability of newer drugs for treating constipation has led to the emergence of scientific evidence surrounding the use of magnesium oxide, which is convenient to administer, of low cost, and safe. Although RCTs have recently shown that magnesium oxide is safe and efficacious for treating constipation, evidence of efficacy for treating symptoms of irritable bowel syndrome, especially the constipation-predominant subgroup, needs to be urgently established. Risk factors for developing hypermagnesemia have been clarified and evidence suggests that appropriate monitoring for this potential side effect is necessary. To be specific, patients with renal impairment of CKD grade G4 or higher and patients who take 1000 mg of magnesium oxide or more daily should be monitored monthly at the time of drug introduction, and monitoring of serum magnesium is recommended in parallel with renal function even during the stable period [6]. However, there is still insufficient evidence to enable comparisons to be made between various laxative drugs and to enable correct prescribing decisions to be made. Drugs such as magnesium oxide are still prescribed based on empirical knowledge, and an accumulation of systematic evidence is still needed (Figure 3).

Chronic constipation is normally treated by general practitioners rather than gastroenterologists. The establishment of systematic, scientifically presented guidelines for treating constipation, which clearly define the position of general practitioners and gastroenterologists and are based on sufficient evidence, are highly desirable.

## Figures and Tables

**Figure 1 nutrients-13-00421-f001:**
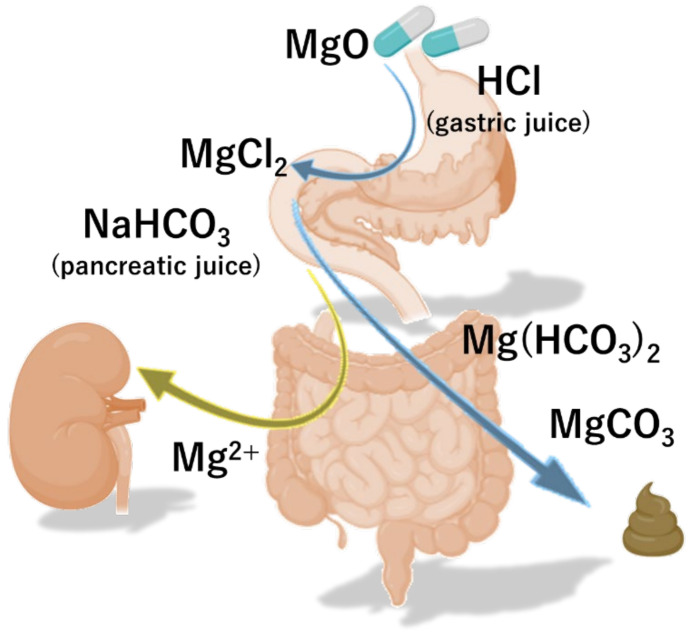
The pharmacokinetics of magnesium oxide. Magnesium oxide (MgO) is converted to sodium hydrogen carbonate (NaHCO_3_) and magnesium carbonate (MgCO_3_) by gastric and pancreatic juice, and exerts its effect as a salt laxative.

**Figure 2 nutrients-13-00421-f002:**
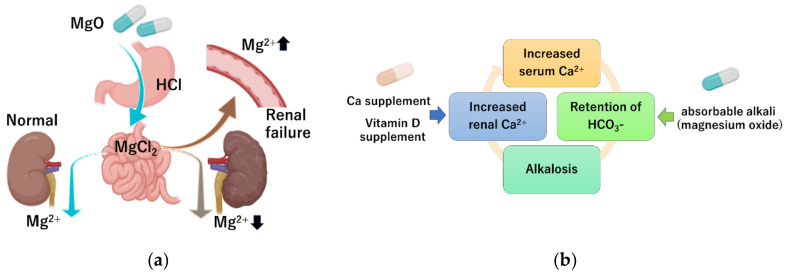
Mechanism of hypermagnesemia and milk–alkali syndrome. (**a**) In patients with impaired renal function, poor magnesium excretion increases the risk of hypermagnesemia. (**b**) Simultaneous intake of excessive calcium and magnesium oxide, which is a non-absorbable alkali, can increase the serum calcium concentration.

**Figure 3 nutrients-13-00421-f003:**
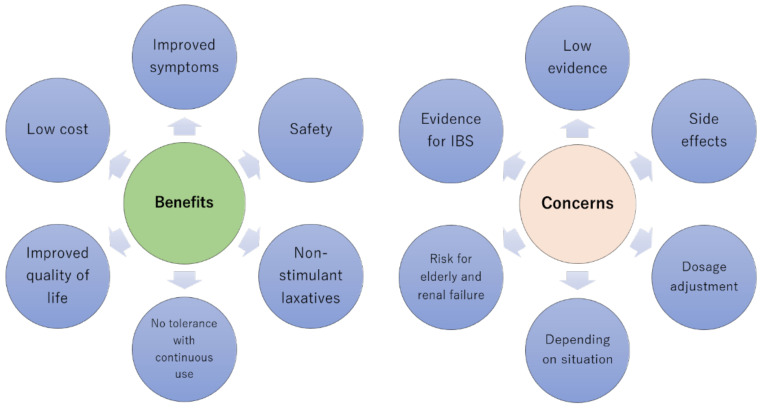
The advantages and disadvantages of magnesium oxide therapy. IBS, irritable bowel syndrome.

**Table 1 nutrients-13-00421-t001:** Comparison of laxative drugs by cost.

	Daily Dose	Cost Per Day (JP)	Cost Per Day (NL)
Magnesium oxide	2 g	€0.18	NA
Magnesium hydroxide	2 g	€0.36	€0.16
Senna	24 mg	€0.09	€0.25
Polyethylene glycol	10 g	€1.02	€0.58
Psyllium seed	7 g	NA	€0.45
Lubiprostone	48 μg	€1.93	NA
Linaclotide	0.29 mg	€1.44	€1.87
Elobixibat hydrate	10 mg	€1.68	NA

Abbreviations: JP = Japan, NL = the Netherlands. NA = not applicable. Data was obtained from the following sites. https://www.kegg.jp/kegg/medicus.html (JP), https://www.farmacotherapeutischkompas.nl/ (NL), 1 euro = 125 yen (December, 2020).

**Table 2 nutrients-13-00421-t002:** Guidelines on the use of magnesium precautions for treating functional constipation.

Institution	Recommendation	Class	Level of Evidence
European Society of Neurogastroenterology and Motility [46]	None	NA	NA
American Gastroenterological Association [38]	Magnesium hydroxide and other salts improve stool frequency and consistency. Absorption of magnesium is limited, and these agents are generally safe. However, there are a few case reports of severe hypermagnesemia after use of magnesium-based cathartics in patients with renal impairment.	NA	NA
American College of Gastroenterology [45]	There is insufficient data to make a recommendation about the effectiveness of magnesium hydroxide in patients with chronic constipation.	No recommendation	Moderate
Asian Neurogastroenterology and Motility Association [39]	Milk of magnesia (magnesium hydroxide) is an osmotic laxative by which the poorly absorbable magnesium ions cause water to be retained in the intestinal lumen. The evidence for its efficacy from randomized control trials is limited.	NA	NA
The Japanese Society of Gastroenterology [37]	Osmotic laxatives are useful and recommended for use in chronic constipation. However, regular magnesium measurement is recommended when using salt laxatives containing magnesium.	Strong	High
The Korean Society of Neurogastroenterology and Motility [40]	Magnesium salts improve stool frequency and consistency in patients with normal renal function.	Strong	Low
An expert panel in Hong Kong [47]	None	NA	NA
Mexican Association of Gastroenterology [41]	Magnesium salts are useful in patients with acute constipation associated with immobilization and should not be used chronically because they produce hypermagnesemia, especially in patients with kidney failure.	Weak	Low
The UK’s National Institute for Health and Clinical Excellence [48]	Substitute a stimulant laxative singly or in combination with an osmotic laxative such as lactulose if polyethylene glycol 3350 plus electrolytes are not tolerated.	NA	NA
The German Society for Digestive and Metabolic Diseases, the German Society for Neurogastroenterology and Motility and the German Society for Internal Medicine [44]	Saline laxatives, such as magnesium hydroxide, are not recommended for chronic constipation because of possible adverse effects.	No recommendation	NA
The Italian Association of Hospital Gastroenterologists and the Italian Society of Colorectal Surgery [42]	The use of magnesium hydroxide is supported by case-series level of evidence.	Weak	Low
French Society of Gastroenterology [43]	The first-line therapeutic interventions recommended by the guidelines are osmotic laxatives (macrogol, lactulose, or milk of magnesia) and bulk-forming laxatives.	Moderate	Moderate

Abbreviation: NA = not applicable.

## Data Availability

Data sharing not applicable.

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
