# Peer review of "Magnesium Oxide in Constipation"

_nutrients, 2021, doi:10.3390/nu13020421_

Round 1
Reviewer 1 Report
This paper is well acceptable. I just have a couple of minor comments and suggestions
L59: prefer „from pancreatic secretion“
L62: prefer „stimulates intestinal motor activity “
L132: A short word on the possible merits of Mg may be appropriate. Mechanistically, opioid receptor antagonists may be advantageous, as opioids act on receptors in both small and large intestine and, second, dampen neuronal activity also in the gut. Relevant may also be the survey by Coyne and coworkers (https://pubmed.ncbi.nlm.nih.gov/24904217/)
L167 Adding „only“ would make more sense - otherwise the „although“ makes no sense.
L193: reference?
L206: recommendation for a sufficient time interval?
L209: Mg will not increase the pH in the entire „gastrointestinal tract“
L217: here and below: The role of Mg in the milk alkali synrome could be more explicitly stated. So far, it is just about calcium
L267: It remains unclear, why the bioavailability of magnesium oxide should be less than that of MgCl2, if it is converted to MgCl2 early on.
L311: A recommendation for what is an appropriate monitoring (for example interval) would be expected
Figure 3: what does the (a) underneath stand for?
Author Response
Reviewer 1
This paper is well acceptable. I just have a couple of minor comments and suggestions
- L59: prefer „from pancreatic secretion“
- L62: prefer „stimulates intestinal motor activity “
Reply: The phrase was changed properly.
L132: A short word on the possible merits of Mg may be appropriate. Mechanistically, opioid receptor antagonists may be advantageous, as opioids act on receptors in both small and large intestine and, second, dampen neuronal activity also in the gut. Relevant may also be the survey by Coyne and coworkers (https://pubmed.ncbi.nlm.nih.gov/24904217/)
Reply: This is a good suggestion based on actual pharmacological action and clinical data. We added these contents.
L167 Adding „only“ would make more sense - otherwise the „although“ makes no sense
Reply: The phrase was added properly.
L193: reference?
Reply: A reference was added.
L206: recommendation for a sufficient time interval?
Reply: Considering the digestion time in the stomach, an interval of at least 2 hours is recommended (Moore JG, et al. Digestive diseases and sciences. 1981;26(1):16-22). We added this point.
L209: Mg will not increase the pH in the entire „gastrointestinal tract“
Reply: This was changed to “the intragastric pH”.
L217: here and below: The role of Mg in the milk alkali synrome could be more explicitly stated. So far, it is just about calcium
Reply: The details how magnesium oxide increases the risk of milk-alkali syndrome have been shown in the next section. Then, we added some remarks in this paragraph.
L267: It remains unclear, why the bioavailability of magnesium oxide should be less than that of MgCl2, if it is converted to MgCl2 early on.
Reply: The low bioavailability of magnesium oxide may be related to its low solubility in water (Lindberg JS, et al. Journal of the American College of Nutrition. 1990;9(1):48-55). We added this point to the paragraph.
L311: A recommendation for what is an appropriate monitoring (for example interval) would be expected
Reply: Thanks for the comment. We added a recommendation as follows:
“To be specific, patients with renal impairment of CKD grade G4 or higher and patients who take 1000 mg of magnesium oxide or more daily should be monitored monthly at the time of drug introduction, and monitoring of serum magnesium is recommended in parallel with renal function even during the stable period (6).”
Figure 3: what does the (a) underneath stand for?
Reply: Thanks for the advice. It is a just mistake. We deleted it.

Reviewer 2 Report
In this paper the Authors analyze the efficacy of Magnesium oxide in constipation. It is a debated and interesting topic about a common condition in HD patients. A comprehensive and extensive literature review of the NCBI database PubMed was also carried out. The article was well conducted and it is interesting in its fields. It is a well-structured paper, written in good English and the References are up dated.
Minor issues:
In the “discussion” section I suggest to better analyze the diagnostic process of constipated patients. In fact, a proper diagnostic process should combine, in addition to clinical and instrumental values, several clinical/physiatric parameters such as puborectalis muscle function, perineal defense reflex, agonist and antagonist muscle synergies, and last but not least, postural examination (lumbar lordosis) and respiratory function.Therefore the following papers should be considered:
“Clinical and instrumental parameters in patients with constipation and incontinence: their potential implications in the functional aspects of these disorders.Brusciano L, Limongelli P, del Genio G, Rossetti G, Sansone S, Healey A, Maffettone V, Napolitano V, Pizza F, Tolone S, del Genio A. Int J Colorectal Dis. 2009 Aug;24(8):961-7. doi: 10.1007/s00384-009-0678-2. Epub 2009 Mar 7..”
“An imaginary cuboid: chest, abdomen, vertebral column and perineum, different parts of the same whole in the harmonic functioning of the pelvic floor.Brusciano L, Gambardella C, Tolone S, Del Genio G, Terracciano G, Gualtieri G, Schiano di Visconte M, Docimo L. Tech Coloproctol. 2019 May 7. doi: 10.1007/s10151-019-01996-x.”
Author Response
Reviewer 2
In this paper the Authors analyze the efficacy of Magnesium oxide in constipation. It is a debated and interesting topic about a common condition in HD patients. A comprehensive and extensive literature review of the NCBI database PubMed was also carried out. The article was well conducted and it is interesting in its fields. It is a well-structured paper, written in good English and the References are up dated.
Minor issues:
In the “discussion” section I suggest to better analyze the diagnostic process of constipated patients. In fact, a proper diagnostic process should combine, in addition to clinical and instrumental values, several clinical/physiatric parameters such as puborectalis muscle function, perineal defense reflex, agonist and antagonist muscle synergies, and last but not least, postural examination (lumbar lordosis) and respiratory function. Therefore the following papers should be considered:
“Clinical and instrumental parameters in patients with constipation and incontinence: their potential implications in the functional aspects of these disorders.Brusciano L, Limongelli P, del Genio G, Rossetti G, Sansone S, Healey A, Maffettone V, Napolitano V, Pizza F, Tolone S, del Genio A. Int J Colorectal Dis. 2009 Aug;24(8):961-7. doi: 10.1007/s00384-009-0678-2. Epub 2009 Mar 7..”
“An imaginary cuboid: chest, abdomen, vertebral column and perineum, different parts of the same whole in the harmonic functioning of the pelvic floor.Brusciano L, Gambardella C, Tolone S, Del Genio G, Terracciano G, Gualtieri G, Schiano di Visconte M, Docimo L. Tech Coloproctol. 2019 May 7. doi: 10.1007/s10151-019-01996-x.”
Reply: Thanks for the suggestion. We agree with the importance of the diagnostic process of constipated patients in terms of which drug to choose. We added sentences and references for the reason.
